# High-Performance Work System, Work Well-Being, and Employee Creativity: Cross-Level Moderating Role of Transformational Leadership

**DOI:** 10.3390/ijerph16091640

**Published:** 2019-05-10

**Authors:** Rentao Miao, Yi Cao

**Affiliations:** School of Labor Economics, Capital University of Economics and Business, 121 Zhangjialukou, Fengtai, Beijing 100070, China; 13126930866@163.com

**Keywords:** high-performance work system, work well-being, employee creativity, transformational leadership

## Abstract

Under the new normal, the economic development mode and growth momentum of China has brought about fundamental changes, which means that the development of enterprises has gradually shifted from being factor-and investment-driven to being innovation-and talent-driven. As the foundation of corporate innovation, employee creativity plays an important role in this process. In the field of strategic human resource management, high-performance work system is the embodiment of its core competence. Although some research has begun to try to explore the impact of high-performance work system on employee creativity, the underlying mechanism and the boundary condition is not yet fully understood. According to the Job demands-resources (JD-R) model, this study theorized and examined whether and when high-performance work system stimulate employee creativity. Using a sample of large and medium-sized enterprises in China, we collected data, which are time-lagged and multilevel, from 266 employees in 61 departments. Results of the hierarchical linear model found that (1) High-performance work system is positively related to employee creativity; (2) High-performance work system positively affects employee work well-being; (3) Work well-being positively affects employee creativity; (4) Employee work well-being partially mediates the relationship between high-performance work system and creativity; (5) Transformational leadership, which represents an important contextual variable in the workplace, moderates the relationship between work well-being and employee creativity; (6) Moreover, we have also revealed that transformational leadership can moderate the indirect effect of high-performance work system on employee creativity. We discussed the theoretical and practical implications of these findings.

## 1. Introduction

With China’s economy entering the new normal, innovation-based strategic human resource management has helped China’s economic development mode and growth engine undergone fundamental changes, which means that the development of enterprises has gradually shifted from factor- and investment-driven to innovation- and talent-driven. As the foundation of enterprises innovation, it is crucial to motivate employee potential and unleash their creativity, which is the key to achieve this transformation [1]. Employee creativity is the generation of novel and useful ideas [2], and it is the cornerstone of organizational innovation [3]. Therefore, how to improve employee creativity effectively has become one of the research purposes.

Unexpectedly, although the human resource management system could effectively improve employee’s motivation, attitude, knowledge, skill, and ability and further affect employee creativity [4], study on the relationship between human resource management system and employee creativity is scarce [3], because most of the studies published in recent years have focused on the role of personality (e.g., Emotional intelligence [5]; proactive personality [6]), organizational situational (e.g., working climate [7]; organizational culture [8]), and leadership (e.g., transformational leadership [9]; moral leadership [10]; abusive leadership [11,12]) as the antecedent variables of employee creativity. It should be noted that some studies use single human resource practice (e.g., pay for performance) to predict employee creativity, and further explore the role played by the mediation mechanism [13]. However, strategic human resource management has always emphasized the systematic perspective that we should use the high-performance work system to study [14]. High-performance work system is a set of human resource practices that are complementary internally (mutually reinforcing human resource practices) and aligned with the organizational strategy externally, aiming at improve employee’s attitudes, behaviors, and abilities, and further improving the competitive advantages of organizational [15]. In fact, compared to other factors, good institutional construction is probably more effectively and significantly in predicting employee creativity [4], since human resource management system is the design and implementation of organization’s philosophies, policies, practices, and processes [16], this view fully reflects the organization’s desire to establish a good system, atmosphere and encourage employees to come up with new and useful ideas to facilitate organizational innovation. In addition, in the work situation, the institution forms the norm, then the norm could, on an overall and long-term basis, guide the employee’s behavior. Therefore, we will explore the relationship between a high-performance work system and employee creativity.

How to achieve well-being has always been the ultimate goal of human to pursuit. Under the new economic normal, the Chinese government pointed out that people’s well-being is as important as the sense of gain and security. This view not only reflects China’s fundamental position on being “people-centered” (“employee-centered” in enterprise), but also corresponds with the point that the development of the enterprise needs to take into account the interests of both parties (enhance the employees’ well-being and the stimulate their creativity) in the new era. Some research topics related to well-being include living well-being, work well-being, and psychological well-being [17]. Among them, the sense of work well-being as a general term for the well-being experience of employees in the work process has gradually attracted the attention of scholars [18]. Current related research indicates that employees’ well-being can be used as a mediator to explain the process mechanism of human resource management, Renee (2008) [19] found that human resource management practices can improve employees’ well-being in the work process and further help to improve their work performance. Nowadays, most countries have emphasized the importance of innovation, including China; in particular, some research has shown that creativity may prompt innovation to enhance company competitiveness [20,21,22]; therefore, creativity has become the proximal outcome of high-performance work system [23]. At present, the research about the formation mechanism of creativity is mainly based on the “componential model” of creativity, some studies explore the mediation effect of intrinsic motivation from the perspective of the psychological [4,13] in the influence process of the human resource system. As has been established, work well-being, which is also the psychological process and similar to the employee’s intrinsic motivation, so we want to know whether it could be a key process variable in the relationship between high-performance work system and employee creativity? Besides, most studies related to a high-performance work system chose a resource-based view [24], social exchange theory [25], AMO (Ability, Motivation, Opportunity) theory [26], and human capital theory [27] to explore the process mechanism of human resource management, but unfortunately, no research has attempted to explore the formation mechanism of the human resource system and employee creativity from the perspective of employee work well-being thus far. Therefore, on the one hand, it is necessary to explore the influence of work well-being as a process mechanism between the human resource system and creativity, because we want to know whether an organization can improve employee work well-being, then work well-being can enhance the employee creativity in turn. On the other hand, constructing strategic human resource management theory from the perspective of work well-being can help us broaden our view about the process mechanism model of a high-performance work system, which takes into account the interests of both employers and employees. Based on above, we think explore the relationship between a high-performance work system and employees’ creativity from the perspective of employee work well-being has great significance, both in a theoretical and in a practical sense.

In addition, although employees’ work well-being reflects the positive and pleasant mental state of employees in work [28], it is not known whether employees can improve their creativity while gaining a sense of work well-being under any condition. In other words, based on the continuing call for contextualization of management research [29], we believe that the positive relationship between employee work well-being and creativity has boundary conditions. We argue that, to a large extent, this relationship depends on the behaviors of the leader in work. Most of the studies show that, the subordinates’ perception of the work environment, the style or traits of the transformational leadership are most likely to affect the further change of positive psychological state (such as work well-being) of the subordinates [30,31,32]. Especially in China, where perceptual thinking is strong, the words and deeds of leadership always affect the subordinate’s recognition of the organizational climate. Research has also shown that employees in a team with transformational leadership are more likely to perceive leadership support and encouragement from psychology and action [31], so employees are more likely to show creativity in addition to work well-being. While in a team with weaker transformational leadership, even if the employees have a sense of work well-being, it may cause “Employee Deviance Behavior” due to the negative influence of the leader, thus inhibiting the generation of creativity. Besides, we use the interaction variable, which means interaction between employee work well-being and the transformational leadership of the team (individual and team interaction), to study its impact on employee creativity echoes the research viewpoint that formation mechanism of employee creativity should follow a systematic perspective [33]. Although past studies have focused on transformational leadership [9], strategic leadership [34], and authorized leadership [35] in predicting employee creativity, few studies have used transformational leadership as a contextual factor to explore the boundary condition between work well-being and employee creativity.

In summary, this study will be based on the “new normal” background of the Chinese economy. Firstly, we will explore the whether a high-performance work system could stimulate employee creativity? Second, whether a high-performance work system could improve employee work-wellbeing? Third, whether work-wellbeing could mediate the relationship between a high-performance work system and employee creativity? Fourth, we try to integrate transformational leadership into the research model of a high-performance work system, and test the moderate effect of the transformational leadership on work well-being and employee creativity. Finally, we test the moderated mediation effect of the model, whether transformational leadership could moderate the indirect effect of a high-performance work system on employee creativity?

## 2. Theoretical Background and Hypotheses

### 2.1. High-Performance Work System and Employee Creativity

The research on strategic human resource management stems from the deep expectations of scholars and line managers that the human resource system can improve organizational performance. Since the 1990s, some scholars, such as Arthur (1994) [36], Huselid (1995) [37], and MacDuffie (1995) [38] have demonstrated the positive relationship between human resource management and performance using empirical data. Therefore, a set of well-defined and complementary human resource practice (human resource practice bundle) are more conducive to organizational high performance [24] than a single human resource practice. Scholars have traditionally referred to it as a high-performance work system [37], a high-involvement work system [39], or a high-commitment work system [40]. The mainstream view points out that a high-performance work system can improve organizational performance through improving employee knowledge, attitudes, and skills [41]. However, there is a lack of agreement in the specialist literature about how to accurately define a high-performance work system [36,37] and the human resource practices contained in them [42]. Although western scholars (e.g., Huselid [37]; Delery, Doty [43]) have explored the constitution of a high-performance work system, which include selection, training, performance appraisal, incentive compensation, career development and work safety [44], compared with the East, especially for China, which is undergoing economic and social transformation, this is obviously not enough literature. Since the human resource practices proposed by Western scholars pay more attention to the “commitment-based” practice, but the industrialization of China started late, even if China has already constructed a complete industrial system, the quality of talents is still uneven. Therefore, some traditional Chinese enterprises still need to use management methods which belong to a control style, such as rigorous selection, discipline management, result-based assessment, etc., which are called “control” practice. Therefore, our study will adopt a high-performance work system that combines practices of “commitment” and “control”.

Previously, scholars have explored the impact of a human resource system on different types of outcome variables related to innovation, but these literatures mostly regard human resource practices as organizational-level variables, some scholars explored the positive relationship between human resource system and organizational innovation [45], organizational creativity [46], or product innovation [47]. Other scholars, in researching cross-level, explored the positive relationship between a high-involvement work system [4] or high-commitment work system [40], which is in the organizational-level, and employee creativity, which is in the individual-level. However, if employees do not correctly perceive the formulation and implementation of organizational human resource practices, the impact of a high-performance work system on employees may not produce the “expected results” [48]. In addition, bounded rationality and rational choice theory point out that perceptions are influenced by our background and cognitive limitations, making us interpret some human resource practices in different ways. Consequently, our perceptions make us evaluate a situation in a specific way and influence our behavior outcome [44]. Therefore, we mainly discuss the impact of perception of a high-performance work system on employee creativity in the individual level. AMO theory (Availability–Motivation–Opportunity) can help us explain to the positive impact of a high-performance work system on employee creativity. AMO theory points out that a high-performance work system can generate the positive outcomes through improving an employee’s work abilities, internal motivations, and job opportunities [49]. Research also pointed out that improving employees’ knowledge, skill, behavioral motivation, and participation opportunities can significantly and positively influence employee creativity and innovation behavior [44].

First, high-performance human resource practices, such as rigorous recruitment, help organizations recruit talented people with creative thinking and innovative skills, which directly ensure the potential for employees to be creative. Extensive training and performance appraisal can further improve the knowledge and skills which are required for an employee to be creative through unified learning improvement and an employee’s developmental performance feedback [40]. Research has shown that the knowledge and skills of employees are an important guarantee for their creativity [50]; Second, some “commitment-based” practices, such as information sharing, incentive compensation and employee engagement, can enhance the intrinsic motivation of employees to be creative [40], such as information sharing, incentive compensation and employee engagement, these practices are not only conducive to constructing the trust relationship between organization and employee [25], but can also help to improve the work climate in terms of employee knowledge sharing, initiative to participate in decision-making, and taking the risk of trying to solve problems with new technologies; Finally, in addition to the required ability and motivation, the effectivity of employee creativity is also linked to the organization’s innovative platform or innovative opportunities, such as employee empowerment, teamwork, and flexible work design, which can provide a platform and opportunity to help employees to translate knowledge, skills and motivation into concrete creative practices [46]. Therefore, human resource practices that provide opportunities for innovation are necessary to motivate employee creativity.

At present, only a few studies have begun to explore the relationship between human resource systems and employee creativity or innovation behavior, such as Escribá-Carda, Balbastre-Benavent and Canet-Giner (2017) [44] using Spanish university data, taking exploratory learning as a mediation variable, to explore the impact of employees’ perceptions of a high-performance work system on innovation behavior; the results show that employees’ perceptions of a high-performance work system can enhance employee innovation behavior through exploratory learning. Furthermore, Liu et al. [3] explored the impact of the interaction between different oriented subsystems in a high-performance work system on employee creativity, the mediating role of employee work skills, and the moderating role of corporate ownership in the model, and their results found that the interaction of performance-oriented human resource systems (such as skills improvement and career development) with the maintenance-oriented human resource systems (such as job safety and workplace quality) has a positive impact on employee creativity.

**Hypothesis 1** **(H1).**
*A high-performance work system positively affects employee creativity.*


### 2.2. High-Performance Work System and Work Well-Being

As an important research topic of positive psychology, work well-being refers to job-related satisfaction, job-related tension, and job-related depression [51]. Current research on work well-being mainly contains the two philosophical orientations, one is the “hedonic approach” that expresses emotional expression [52,53], the other is the “eudaimonic approach” that reflects the meaning of work, work value and personal development [54,55]. Scholars that belong to organizational behavior and human resources have formed three research perspectives of work well-being based on the above two philosophical orientations: subjective well-being perspective, psychological well-being perspective, and integrated well-being perspective. Among them, subjective well-being is based on the hedonic approach, including positive influence, negative influence and life satisfaction [52]; psychological well-being perspective is based on the eudaimonic approach; the most common theoretical model consists of six dimensions, including autonomy, environmental mastery, personal growth, positive relations with others, purpose in life, and self-acceptance [56]. The integrated well-being perspective combines subjective well-being with psychological well-being; this view argues that in the research of work well-being, the perspective of subjective well-being and psychological well-being are indispensable. At present, the perspective of integrated well-being perspective is gradually favored by researchers, who define work well-being as “overall quality of individual experience and effectiveness at work” [57]. Consistent with the perspective of integrated well-being perspective, work well-being in this study also considers both subjective well-being and psychological well-being. In an increasingly competitive business environment, work well-being, which is a positive psychological state of employees, has gradually become one of the key factors in affecting organization competitive advantage [58]. Therefore, how to effectively help employees improve their work well-being has become a management problem that the organization needs to solve.

At present, there are two competing perspectives about the impacts of a high-performance work system on employee work well-being—the unitarist perspective and the pluralist perspective. The unitarist perspective is aimed at “win–win”(employer and employee); this perspective tends to think that the organization’s goals are consistent with the interests of employees, and it overlooks or downplays the conflict of interest between employers and employees [18], especially when the organization implements a high-performance work system to enhance employees’ abilities, motivation, and opportunities to perform, employees are more likely to attribute the intention of a high-performance work system as caring about and developing employees to better contribute the organizational goals [59]. For example, Boxall and Macky (2014) [60] found that human resource practices such as autonomy, information sharing, compensation, and training development can influence (improve or reduce) employee work well-being through improving employee job satisfaction or reducing work-life balance satisfaction. Further, Huang et al. [61] pointed out that employees can acquire and maintain work-related resources from a high-performance work system, and that they are therefore more likely to perceive higher degree of well-being, job satisfaction, and better performance. Conversely, the pluralistic perspective does not consider organizational goals to be consistent with employee interests [62], especially when organizations pursue short-term benefits or employers seek personal interests at the expense of employee benefits [59]. Therefore, the perspective of pluralistic believes that human resource practice has a negative impact on employee work well-being. For example, some scholars found that the human resource systems, which includes recruitment, training development, performance appraisal, compensation, employee participation and communication, could significantly reduce employee job satisfaction and emotional commitment, and increase work pressure, so that the human resource systems will reduce work well-being significantly [63]. In addition, some research has also found that human resource systems (skill enhancement, job incentives, empowerment and family-friendly policies [64]; recruitment, career development, compensation, performance evaluation, job design, employee engagement and communication [65]) can reduce employee work well-being by increasing employee anxiety, role overload and emotional exhaustion. From here, we see that Western scholars have not reached a consensus on the conclusions of the high-performance work system on employee work well-being. However, it can be seen that the studies above present two characteristics: first, the research findings are based on western management situations; second, the human resource practices adopted are belong to the “commitment-based” practices. At the same time, we also found that the opposite conclusions are contrary to our initial understanding (especially for China)—the “commitment-based” practices, which draw on signal theory and are “employee-centered”, should be conducive to employee work well-being. Nevertheless, unlike the western commitment-based practice, based on the Chinese management context (Traditionally, Chinese people have a tendency to think that “not to worry about poverty, but rather about the uneven distribution of wealth” (The idiom comes from “The Analects”, which is a traditional book in China and has affected Chinese people deeply. The "uneven" here is not a simple imbalance, but everything in its right place, which means everyone gets their share under a fair distribution system. At the same time, this view is also considered to be in line with the idea of socialism’s fairness and justice), and Chinese people have always been pursuing the idea that “employee-centered” thinking); therefore, how does the relationship between the high-performance work system that takes into account both “commitment-based” practices and “control” practice and the employee work well-being?

Drawing from the job demands–resources (JD-R) model, we theorize that a high-performance work system will lead to employee work well-being by increasing the availability of psychological and social resources. The JD-R model [66] divides the work environment into two broad categories: job demands and job resources. Job demands refer to physical, psychological, social, or organizational demands resulting in physical and psychological costs; Job resources refer to physical, psychological, social, or organizational resources contributing to the achievement of goals and reducing adverse physical and psychological consequences (meet the “employee-centered”). JD-R model has three core assumptions: one is the “two path” Hypothesis—the impact of work on employees is achieved through two paths: loss (job demands effect) and gain (job resources effect); second, “buffering” assumption—job resources can buffer or reduce the loss and negatives impact of high-intensity job demands on employees; Third, the “response” Hypothesis—job resources can improve employee motivation and job involvement under high-intensity job demands, which means that high-intensive job demands will motivate employees to make full use of high-level job resources, and further work in a more positive state and achieve goals.

We can find that the high-performance work system that takes into account both control and commitment-based practices in the Chinese management context is similar to the third Hypothesis in the JD-R model—the “response” Hypothesis. Specifically, high-intensity job demands are similar to control-based practices that include rigorous selection, results-based performance appraisal, and strict discipline management, while the high-level job resources are similar to commitment-based practice, involving extensive training, employee engagement, information sharing, rewards. Traditionally, Chinese people have always had a thinking tendency to “not worry about poverty, but rather about the uneven distribution of wealth”. When it is in the high-intensity work, the control-based practices are in line with the traditional values—make no exception to fairness and justice. Chinese people pay attention to these values (such as rigorous selection, which means fair selection; results-based performance appraisal, which means distributive justice; strict discipline management, which means procedural justice). In addition, commitment-based practices are in line with the “employee-centered” approach. In this way, employees can better convert sufficient job resources into high-level motivations, and then engage in high-intensity job involvement; in turn, such high-intensity involvement will generate more high-level resource gains and further form a virtuous circle; finally, these paths will enhance employee work well-being. Empirical study has also found that high-intensity job demands and high-level job resources have the most significant incentive effect on employees compared to low-intensity job demands and high-level job resources [67]. Therefore, we believe that in the context of management in China, when organizations implement high-intensity control-based practices (job demands), high-level commitment-based practices (job resources) will become an effective complement to control-based practices and form an “complementary effect”, which will significantly stimulate the motivation of employees, and promote job involvement, improve job satisfaction and reduce burnout, and further enhance their work well-being.

**Hypothesis 2** **(H2)**.
*High-performance work systems positively affect employee work well-being.*


### 2.3. Work Well-Being and Employee Creativity

Currently, scholars have discussed the relationship between work well-being and work performance, relationship performance, safety, intention of turnover, customer’s satisfaction, and organizational performance in the individual/organizational level [68,69], but few studies have focused on the impact of work well-being on employee creativity. Componential theory of creativity points out that creativity is closely related to employees’ intrinsic motivation, domain-relevant skills and creativity-relevant skills [50].

Attribution models of motivation can provide a clue for building a relationship between work well-being and employee creativity [70]. The model argues that compared with the employees who lack happiness, thus attributing failure to themselves (attribution of internal subjective), the employees who are happy tend to attribute failure to the temporary disadvantage of the situation (attribution of external objective) [70]. As the study found, some of the disadvantages in the organization are more likely to have a negative impact on employees with low work well-being than those with high work well-being; on the other hand, compared to employees with low work well-being, some of the favorable conditions are more likely to have a positive impact on employees with high work well-being, thus can affect employee creativity [71].

In addition, employees with high happiness will be more extroverted and outgoing. While those with low happiness will be shier and more introverted, especially sometimes these employees tend to use controversial interpersonal strategies to interact with others and even behave acrimoniously [70], which in turn can cause antagonism amongst colleagues. Therefore, compared with happy employees, employees who lack happiness will feel less support from their supervisors and colleagues [72], thus lacking good communication, information sharing and necessary resource support. Therefore, employees with low happiness are not conducive to the improvement of their domain-relevant skills and creativity-relevant skills.

Moreover, compared with happy employees, those who lack happiness argue that they lack control over what happens in work, always feel things that far exceed their ability. Besides, these less happy people are even sometimes pessimistic about the future [73]; we argue that this sense of frustration can further affect and limit the perception of employees who lack happiness, causing them to lose the initiative and enthusiasm to deal with problems, and finally reduce their motivation to work. In contrast, employees with high work well-being are satisfied with their work, self-value and psychological state, and they are good at winning the goodwill of supervisors and colleagues through good interpersonal strategies, which will help them to get support, thus increasing their domain-relevant skills and creativity-relevant skills finally; At the same time, employees with high work well-being will also broaden strategy of information processing and explore their cognitive potential, which will help them to improve the creative initiative and motivation to deal with problems.

**Hypothesis 3** **(H3)**.
*Work well-being positively affects employee creativity.*


### 2.4. The Mediating Role of Work Well-Being

Although some studies have explored the mediation mechanism between a high-performance work system and employee creativity, such as domain-relevant skills [3], exploratory learning [44], we expect a high-performance work system to be positively related to employee creativity in light of the JD-R model; one possible variable is work well-being. Drawing from the JD-R model, a high-performance work system that combines commitment-based and control-based practices provide employees with “high-high” job environment (job demands and job resources) to meet the needs of acquiring work well-being, which in turn fosters opportunities and resources for employee creativity. Especially in China, in contributing to the organization for the high-performance work system, employees will perceive a greater sense of justice (from control-based practices) and “employee-centered” (from commitment-based practices), thus promoting the employees work well-being. Work well-being can be conducive to improving employees’ intrinsic motivation, domain-relevant skills, and creativity-relevant skills, and therefore, employee creativity will be enhanced. Specifically, in keeping with the attribution models of motivation [74], work well-being facilitates employee’s positive state in psychology, leading to employee creativity improvement. As mentioned previously, the sense of justice and “employee-centered” (fostered by a high-performance work system), as manifested in work well-being, constitute a mediating mechanism for the documented a high-performance work system–employee creativity relationship.

**Hypothesis 4** **(H4)**.
*Work well-being mediates the positive relationship between a high-performance work system and creativity.*


### 2.5. The Moderating Role of Transformational Leadership

Apart from internal elements such as intrinsic motivation, domain-relevant skills and creativity-relevant skills (individual factors), employee creativity also depends on the support (situational factors) of external elements [50]. Thus, the impact of work well-being (internal elements) on employee creativity has boundary condition. Leadership as an important contextual factor in the workplace and will undoubtedly have a significant impact on employee creativity [9]. Among the leadership styles, transformational leadership, which is an effective and high-profile leadership style, plays a critical role in the research of creativity [75]. Bass (1985) [76] suggested that transformational leadership is a four-dimensional concept, including charisma or idealized influence, inspirational motivation, individualized, consideration, intellectual stimulation. However, based on Chinese specific situation, scholars Chaoping and Kan (2008) [77] pointed out that transformational leadership is a four-factor construct, which includes articulate vision, charisma, moral modeling, and individualized consideration; and they further suggested that articulate vision and charisma are similar to those in the West, but for individualized consideration, western leader pay more attention to employees work category, while Chinese leaders pay more attention to employee family and life besides work itself; The difference in this concept is moral modeling, which reflects the cultural background in China; Confucius believed that fostering individual’s personality and virtue are the foundation of a society, and the moral standards of leader have a significant impact on employee behavior in work place. Currently, although most studies have demonstrated the positive impact of transformational leadership on employee creativity [9,75], unlike previous studies, drawing from the “interactive point of creativity” (interaction of work well-being and transformational leadership) [78], we use systematic perspective to consider individual and contextual factors, which can influence the creativity, and believe that transformational leadership can moderate the effect of work well-being on employee creativity.

Specifically, employees who are affected by high-level transformational leadership can easily perceive the support, care, and help of leaders from articulate vision, charisma, moral modeling, and individualized consideration. For example, (1) ***articulate vision*** transformational leadership will motivate employees through drawing a grand and clear vision to employees, or express their expectation for employees through providing employees with valuable and challenging work, thus making employees willing to go beyond individual interests and strive for team interests. In this way, employees are more likely to pay attention to the task; (2) ***charisma*** transformational leadership care and help employees in their work and life, and also create opportunities for employees to show their talent; (3) ***moral modeling*** transformational leadership will set moral example for employees and become a role model, which could be identified, respected, and trusted by employees; (4) ***individualized consideration*** transformational leadership attaches importance to the personal development of employees, especially to employees’ need of achievement and growth. Besides, in order to enable employees to have higher ability to complete the task, transformational leadership also encourages employees to learn the domain-relevant skills. Therefore, it can be found that transformational leadership provides the necessary motivation and skills for employees to complete their tasks. In fact, empirical research has shown that internal motivation [79] and domain-relevant skills [3] are important to stimulate employee creativity. In addition, employees who are affected by transformational leadership will also create a sense of “well-being and satisfaction”, which could further increase their trust, loyalty, and belongingness to the organization and leader (In China, to a large extent, the recognition of leadership is largely equivalent to the recognition of the organization.), on the other hand, these employees will also realize their status as “owners”, which could make it easier for task motivation of creativity. Therefore, we will suggest that employees who have a sense of work well-being, which is a positive psychological state to stimulate employee creativity (individual factors) are more likely to show their creativity if they are supported or helped by transformational leadership (situational factors). Thus, based on the interactive point of creativity [78], when the transformational leadership is higher, the positive relationship between work well-being and employee creativity is stronger.

Conversely, employees who are affected by low-level transformational leadership will have doubt and confusion about organization and leader, and sometimes even begin to doubt the positive role of leadership in the organization because they do not perceive the care, support, and help from leader. Although employees have a sense of work well-being, which can be seen as a prerequisite for stimulating creativity, low-level transformational leadership makes employees ambiguous about the organization’s vision and goals, and at the same time, employees also do not perceive the moral role and support from leaders, thus their sense of identification and commitment for organization will gradually decrease, and sometimes even begin to doubt their motivation of work. In this situation, low-level transformational leadership will inhibit the positive impact of work well-being on employee creativity.

Some similar studies, such as Bono et al. [80], have found that the transformational leadership of superior will strengthen the positive relationship between employee psychology state and work result. Further, Oreg and Berson (2011) [81] pointed out that transformational leadership will weaken the positive impact of employees’ dispositional resistance to change to employees’ resistance intentions. Therefore, we argue that as an important situational variable in the workplace, transformational leadership will moderate the relationship between work well-being and employee creativity.

**Hypothesis 5** **(H5).**
*Transformational leadership moderates the positive relationship between Work well-being and Employee creativity in such way that the relationship is stronger when transformational leadership is high, than when it is low.*


Further, combining the mediation effect with moderation effect, our study can show a more complex theoretical model. Specifically, work well-being mediates the positive relationship between a high-performance work system and employee creativity, but size of mediation effect depends on the level of transformational leadership. Generally speaking, when employees perceive higher level of transformational leadership, the positive relationship between work well-being and employee creativity is stronger, thus the work well-being will more transmit the effect of a high-performance work system on employee creativity. In contrast, when employees perceive lower level of transformational leadership, the positive relationship between work well-being and employee creativity is weaker, thus the effect that a high-performance work system can have on employee creativity will be less transmitted through work well-being.

**Hypothesis 6** **(H6)**.
*Transformational leadership moderates the mediation effect of Work well-being between a high-performance work system and employee creativity (Hypothesis 4) in such a way that the mediation effect is stronger when transformational leadership is high than when it is low.*


The theoretical model of this study is depicted in Figure 1.

## 3. Methods

### 3.1. Participants and Procedure

First, we determined the theoretical model by discussion and made up the questionnaire, and then we sent a message to the CEO or Human Resources Director of some large and medium-sized companies in China, including Beijing, Shanghai, Jiangsu, Shanxi and other regions, involving logistics automation, hotel management, packaging and printing, food processing and intelligent manufacturing industries. We explained our research objectives and survey needs; some of companies responded to our emails and agreed to our request. Next, in these agreed companies, their CEO or HR Director helped us recommend some departments that were willing to participate in the research. It should be noted that all of our participants are voluntary. Our data collection took 2 months and passed 3 time points. In order to avoid common variance, our study adopts the paired questionnaire survey method of “department supervisor-subordinator”. Specifically, the “Supervisor questionnaire” is filled in by the department supervisor, and then the supervisors of each department randomly select 3–8 employees in the department and evaluate their creativity. The “Employee Questionnaire” is filled in by the selected employees and they evaluate the organization’s high-performance work system (i.e., rating human resource practices according to their own experiences), work well-being and supervisor’s transformational leadership. We use the method of questionnaire number to ensure the matching of the supervisors and their employees. At the same time, in the process of filling out the questionnaire, in order to eliminate the doubts of the participants and improve the objective authenticity of the questionnaire, all of our questionnaires were answered anonymously. All questionnaires were distributed on-site and collected on-site. Specifically, the researchers commissioned their friends to conduct questionnaires according to the research design, and sent them back by mail. According to the research design, we collected data at three points in time. First, 500 “Employee questionnaires” were issued at time1, and we ask employees only evaluate their perception of a high-performance work system, 373 valid questionnaires were collected with recovery rates of 74.6%; One month later, the above-mentioned 373 employees were issued “Employee questionnaires” again, this time employees evaluated their well-being and their transformational leadership, 298 valid questionnaires were collected with recovery rates of 79.9%; After another month, we issued questionnaires to the supervisors of the above 298 employees, and we finally identified 68 department supervisors, so we issued “Supervisor questionnaire” to 68 department supervisors, 61 department supervisors participated survey, which corresponds to 266 employees and with recovery rates of 89.2%. In the survey of the paired questionnaires, the maximum number of participants in the surveyed teams was 5, and the minimum number was 3, with an average of 4.70 (SD = 0.639). The average age of all the participants was 29.41 (SD = 6.316). According to statistical analysis, among the 266 participants, there were 113 males and 153 females, accounting for 42.5% and 57.5% of the total number of participants respectively. In terms of education level, the participants below undergraduate accounted for 44%, undergraduate and above accounted for 56%; working years averaged 6.746 years (SD = 6.204); for post, there were 10 senior managers, 13 middle managers, 19 grassroots managers, 205 grassroots employees; company ownership, 87 participants in state-owned enterprises, 179 participants in private enterprises, accounting for 32.70% and 67.30% of the total respectively; for income, there are 173 participants are low earners (below 8000 Yuan), 93 are high earners (above 8000 Yuan), and no extreme value (Shown in Table 1).

### 3.2. Measures

The variables measures (7-point Likert scale) used in this study were either developed in China or showed excellent psychometric properties when used in Chinese. Miao et al. (2014) [15] developed and validated the high-performance work system in Chinese enterprises. Zheng et al. (2015) [17] developed a measure of well-being in the Chinese context and has demonstrated desirable reliability and validity, we selected some of the measurement items about work well-being. Li and Shi (2008) [77] developed a localized transformational leadership scale in China. The scale of employee creativity was developed by Zhou and George (2001) [82]. Since the target samples (well-educated employees and their supervisors) were Chinese, in order to form the Chinese questionnaire, the English items were translated into Chinese by an associate professor (earned a Ph.D. in management from the University of Michigan) in human resource industry and then translated back into English by another assistant professor (earned a Ph.D. in management from the Chinese University of Hong Kong). Finally, two management professors who are bilingual in English and Chinese compared the English version to the original, and concluded that they were highly comparable and made further modifications to the Chinese items to improve accuracy and readability.

***High performance work system.*** Target employees reported their own experience of a High-performance work system rather than an aggregate overall assessment of their firm’s human resource systems according to Miao et al. (2014) [15]. A total of 8 dimensions and 33-item High performance work system measure; sample items were “Our company has a strict selection process (written test, interview, etc.).” “Training content provided by our company is relatively systematic, such as corporate culture, management/professional skills.” “The team is widely used in company.” ”The company will provide more favorable treatment for key core talents.” ”The company encourages employees to share expertise and skills.” Next, we conducted a reliability analysis, Cronbach’s α was 0.878. However, Rwg_(j)_ was 0.632, which was below the conventional aggregation cutoff of 0.70 [83], thereby supporting the standpoint of strategic researchers that meaningful variability exists in employees’ individual experiences of human resource system [3]. In addition, this scale is widely used in the Chinese context, for example, Miao, Zhou, Liu, Pan, and Liu (2015) [84] used it in Chinese article.

***Work well-being.*** Zheng et al. (2015) [17] scale was used to measure target employees’ work well-being in their work areas. Specifically, there are 6-item in this scale. For example, “I am satisfied with my work responsibilities.” “In general, I feel fairly satisfied with my present job” “I find real enjoyment in my work.” “I can always find ways to enrich my work” “Work is a meaningful experience for me” “I feel basically satisfied with my work achievements in my current job”. We conducted a reliability analysis, Cronbach’s α was 0.897. This scale was also used in Chinese context, such as Wang et al. [85] used this scale in their study.

***Transformational leadership.*** Target employees rated their supervisors, for which we chose Li and Shi (2008) [77] scale. Specifically, there are 4 dimensions and 26-item in this scale, for example, “my leader can put his/her personal interests behind the interests of the collective and others.” “my leader often work with employees to analyze the impact of their work on the overall goals of the department”, ”my leader is decent and selfless”, ”my leader can consider the real conditions of employees when contacts with them”, ”my leader can let his/her subordinate know the prospect of the department/unit”. We conducted a reliability analysis, Cronbach’s α was 0.957. Similarly, this Chinese localization scale is also widely used by authors, such as Li et al. [86], and GAO et al. [87].

***Employee creativity.*** Supervisors rated target employees’ creativity using the 13-item scale from Zhou and George (2001) [82]. Sample items were “Employee A will Come up with new and practical ideas to improve performance.” “Employee A will Searches out new technologies, processes, techniques, and/or product ideas.” ”Employee A will exhibit creativity on the job when given the opportunity to.” ”Employee A Often has new and innovative ideas.” ”Employee A is not afraid to take risks.” We conducted a reliability analysis, Cronbach’s α was 0.950. This scale is widely used in studies, such as Liu et al. (2017) [3] and Alzghoul et al. (2018) [88] have used in studies.

***Control variables.*** According to Li and Zhang (2007) [89] and Miao et al. (2014) [15], our study incorporates employee gender, employee age, employee enrollment years, employee education level, employee position and employee income as control variables. Descriptive statistical analysis and correlation analysis were carried out with SPSS22.0 software (IBM, Armonk, NY, USA). Confirmatory factor analysis was performed with AMOS21.0 software (IBM, Armonk, NY, USA). Direct effect test, mediation effect test, moderate effect test and moderated mediation effect test were completed with HLM6.02 software (belong to Scientific Software International, Cambridge, MA, USA).

## 4. Results

### 4.1. Discriminate Validity

In this study, the collected data was subjected to confirmatory factor analysis (CFA) using AMOS21.0, and we use the model validity method to test the discriminate validity of the selected variables, as shown in Table 2.

As can be seen from Table 2, the four-factor model fits very well with the data (*RMSEA* = 0.055, *NNFI* = 0.940, *CFI* = 0.972). The AIC (Akaike information criterion) was used to compare the baseline model (four-factor model) with the candidate model. The difference test between the measurement model and the candidate model shows that the four-factor model is significantly better than the alternative three-factor model, two-factor model and single-factor model. At the same time, by comparing the AIC values (the smaller the values are, the model better), the four-factor model is also superior to the alternative model. Therefore, the above variables have good discriminant validity and are indeed four different constructs.

### 4.2. Convergent Validity

Since the theoretical model constructed in this study involves two levels of team and employee individual, the team level variable is transformational leadership, while high performance work system, work well-being, and employee creativity belong to individual level variables, so we should calculate the Rwg_(j)_ (within-group interrater reliability)and ICC (Intra-class correlation) of transformational leadership, of which two indexes determine whether individual data at the employee level can be aggregated at the team level. The calculated results show that the Rwg_(j)_ coefficient of the transformational leadership is 0.92, which is meet the demand that Rwg_(j)_ in the cross-level study should above 0.70 [83]. At the same time, ICC(1) and ICC(2) of transformational leadership are 0.12 and 0.78 respectively, which also meet the test criteria for ICC(1) > 0.05 and ICC(2) > 0.50 proposed in the cross-level study [90], so we can conclude that the convergent validity of the transformational leadership is good. As a result, transformational leadership at the individual level of employees can be aggregated to the team level.

### 4.3. Common Method Deviation Test

Although this study adopts the paired survey sample of “department supervisor-subordinators” to broaden the sample source channel to a certain extent, in order to avoid the Common Method Deviation of the data, the Harman single factor test is still carried out, and the test results show that the interpretation of the first factor of unrotated is 26.344%, which is consistent with the first factor proposed in the empirical study to explain that the covariate needs to be less than 40% [91]. Then, following the recommendations of Podsakoff et al. (2003) [92], the “potential method factor effect control method” was used to test the common method deviation. The results show that after adding a method factor based on the four-factor model, the five-factor model fits the index: χ^2^(44) = 200. 259, *NNFI* = 0. 930, *CFI* = 0. 966, *RMSEA* = 0. 056. Through the comparison of the five-factor model and the four-factor model, we found that the fit index (χ^2^, *NNFI* and *CFI*) in has a certain degree of decrease, while although the *RMSEA* in four-factor model has a certain degree of increase, it is not more than 0.02. Therefore, it can be judged that the model has not been significantly improved after adding a method factor, indicating that the common method deviation in this study is not serious.

### 4.4. Multicollinearity Test

In addition, we have centralized each variable and found that the tolerance of each variable is between 0.647 and 0.765, and the variance expansion factor VIF (Variance Inflation Factor) is between 1.307–1.546, which is far below the critical value of 10. Therefore, we believe that the model does not exist a serious multicollinearity problem.

### 4.5. Correlation Analysis

Next, we performed some data feature analysis on each variable, the mean (M), the standard deviation (SD), the correlation coefficient (*r*) of the main variables, and the reliability coefficient (α). The results of the analysis are shown in Table 3. We can find that there is a significant correlation between the main variables of the model study. The high-performance work system is positively correlated with work well-being (*r* = 0.635, *p* < 0.01), and is positively correlated with employee creativity (*r* = 0.622, *p* < 0.01), work well-being is positively correlated with employee creativity (*r* = 0.670, *p* < 0.01). Therefore, it is assumed that H1–H3 can be initially verified, which provides the necessary premise for testing the mediation effect of work well-being. In addition, the reliability analysis also showed that the Cronbach’s α coefficients of the four variables were all greater than 0.70.

### 4.6. Hypotheses Testing

#### 4.6.1. The Main and Mediating Effects of Work Well-being

The independent variable of this study is a high-performance work system, and the dependent variable is employee creativity. According to this, the theoretical model is constructed. The method of Baron and Kenny’s (1986) [93] three-step approach is used to examine the mediating effects, specifically: (1) examine whether the effect of a high-performance work system on employee creativity is significant; and (2) examine whether the effect of a high-performance work system on work well-being is significant; (3) If the first two tests are significantly correlated, further examine whether the impact of a high-performance work system on work well-being and employee creativity is significant; if the result finds that the relationship between a high-performance work system and employee creativity is no longer significant or significantly reduced, then this demonstrates the impact of work well-being fully or partially mediating the impact of a high-performance work system on employee creativity.

As shown in Table 4, first, we put the employee gender, employee age, employee enrollment years, employee education level, employee position, and the employee income in model 1 (M1) as control variables. Then, put a high-performance work system in model 2 (M2), and start the regression test. It was found that the high-performance work system had a positive and significant impact on employee creativity (*b* = 0.457, *p* < 0.01), so Hypothesis 1 received support. Next, model 8 (M8) uses work well-being as a dependent variable, and put a high-performance work system into the model, the results show that there was a positive and significant correlation between a high-performance work system and work well-being (*b* = 0.498, *p* < 0.01), so Hypothesis 2 is also supported. In model 3 (M3), the relationship between work well-being and employee creativity in significant (*b* = 0.401, *p* < 0.01), so the Hypothesis 3 received support. Finally, in model 4 (M4), a high-performance work system and work well-being were added to the model for examining, it was found that the coefficient and significance between a high-performance work system and employee creativity was reduced (*b* = 0.288, *p* < 0.01), but the relationship between work well-being and employee creativity was still positive (*b* = 0.318, *p* < 0.01). According to the above statistical analysis, work well-being is partially mediated by the relationship between a high-performance work system and employee creativity, so Hypothesis 4 is confirmed.

In addition, to more accurately examine the mediating role of work well-being between a high-performance work system and employee creativity, we also examined whether the indirect effect (a × b) of work well-being between a high-performance work system and employee creativity is significant. We adopted the method of PROCESS to test mediate effect, the result shows that 95% confidence interval of the indirect value (a × b) (shown in Table 5) is [0.116, 0.288], which does not include 0, so Hypothesis 4 is supported again.

#### 4.6.2. The Cross-level Interaction between Work Well-being and Transformational Leadership

For the test of the moderating effects of Hypothesis 5, as shown in Table 4 model 5 (M5), transformational leadership is put into the model, and it was found that the transformational leadership (*b* = 0.444, *p* < 0.01) is significantly related to employee creativity. Then, we added the interaction variable (WWB×TLS) into the model 6 (M6), and found that the interaction variables’ coefficient (*b* = 0.252, *p* < 0.01) is significant. Thus, Hypothesis 5 is supported.

#### 4.6.3. The Moderated Mediation Model

The mediating role of work well-being and the moderating role of transformational leadership should occur simultaneously. So we use the “Total Effect Moderation Model” to test the moderated mediation effects of model—Hypothesis 6. This kind of method overcomes the shortcomings of separating the mediating effect and the moderating effect in previous studies, and uses a more comprehensive perspective to incorporate the mediating effect and the moderating effect. Edwards and Lambert (2007) [94] argue that at the theoretical level, the moderation effect of the “first stage” or “second stage” can be discussed respectively, but when we analyzed, the “first stage” and the “second stage” were not distinguished. According to Edwards and Lambert’s method, this study constructs the following two equations:Employee Creativity = a_05_ + a_x5_ Work Well-being + a_z5_ transformational leadership + a_xz5_ Work Well-being × transformational leadership + em5(1)
Employee Creativity = b_020_ + b_x20_ High-performance work system + b_m20_ Work Well-being + b_z20_ transformational leadership + b_xz20_ High-performance work system × transformational leadership + b_mz20_ Work Well-being × transformational leadership + e_y20_(2)

Equation (1) refers to the first stage effect, direct effect, and indirect effect, and Equation (2) refers to the second stage effect, direct effect, and indirect effect. Furthermore, the parameters of Table 6 obtained by regression analysis are brought into Equations (1) and (2), and further, using bootstrapping method, the effect of first stage, the second stage, direct, indirect and total are shown in Table 7.

We used 1000 sample self-sampling (Bootstrapping) (see Table 7) to calculate the path coefficient, indirect effects, and significance of difference. We found that in the indirect effect, the coefficient of difference (*γ* difference = 0.066, *p* < 0.05) between high and low is significant (TLS), therefore, the transformational leadership moderates the mediating effect of work well-being, supporting the moderated mediation role of a high-performance work system and employee creativity; therefore, Hypothesis 6 is supported.

Further, we used a simple slope method to draw the diagram of moderating effect. The lower or higher level transformational leadership was obtained according to the average of variables to plus or minus one standard deviation. Figure 2 shows that the relationship between work well-being and employee creativity is stronger when the transformational leadership is higher (*simple slop* = 0.377, *p* < 0.01) than when the transformational leadership is lower (*simple slop* = 0. 311, *p* < 0.01), and the results are consistent with Hypothesis 5. Figure 3 shows that the mediating effect of work well-being is stronger when the transformational leadership is higher (*simple slop* = 0.177, *p* < 0.01) than when the transformational leadership is lower (*simple slop* = 0. 111, *p* < 0.01), and the results are consistent with Hypothesis 6.

## 5. Discussion

### 5.1. Theoretical Implications

First, the result indicates that a high-performance work system that takes into account both control-based and commitment-based practices can have a positive impact on employee work well-being. This conclusion does match the majority of western research about a high-performance work system [60,63], which suggests that the human resource system could decrease the employee work well-being through increasing employee stress, anxiety, role overload and emotional exhaustion. We believe that under the Chinese management context, a high-performance work system that takes into account both control and commitment practices, these factors can be supplementary to each other and be mutually reinforced. We think that this configuration could improve employee work well-being. In addition, we believe that under the Chinese management context, using a systematic perspective to consider the role of control-based and commitment-based practices in increasing employee work well-being can enrich and expand the research about influence factors of work well-being from the perspective of strategic human resource management.

Second, we found that employee work well-being is positively related to the employee creativity, and the impact of a high-performance work system on employee creativity can be achieved through improving employee work well-being. This research design is a new try, new discovery, and new perspective about the formation mechanism of creativity, because most studies in the past focused on intrinsic motivation [3,13] or domain-relevant skills [3] to explore the formation mechanism of employee creativity, while our study pays attention to the mechanism exploration from the perspective of employee work well-being. This mediation result not only indicates that a high-performance work system and employee work well-being are two effective sources of employee creativity, but also shows that a high-performance work system can balance the interests of both employer and employee (i.e., ensure the improvement of employee work well-being and their creativity). Therefore, a high-performance work system can effectively enhance employee creativity by improving employee’ internal work perception and experience, which represents employees’ work meaning and value, autonomy, efficacy and sense of belonging. So, we believe our research enriches and promotes the research on the formation mechanism of employee creativity in the field of strategic human resource management.

Finally, transformational leadership moderates the mediating effect of employee work well-being between a high-performance work system and employee creativity. Thus, we can conclude that the higher the level of transformational leadership, the stronger the employee work well-being and the stronger the employee creativity. This result is also in line with the research statement that “research on employee creativity should be based on a more collaborative perspective [33]”, which means we could explore the mutual combination of different contextual variables (work well-being and transformational leadership). This perspective also reflects a point that the impact of mediation effect of work well-being has boundary conditions. Therefore, it is of great significance to examine the transformational leadership as the boundary condition of the formation mechanism of employee creativity.

### 5.2. Practical Implications

Our study has at least three practical implications: First, in the context of Chinese management, the employee-perceived high-performance work system has a positive effect on the improvement of employee work well-being, especially the high-performance work system that takes into account the job demands, which belongs to control-oriented practice, and job resources, which belongs to commitment-oriented practice, and is more suitable for the improvement of employee work well-being in the Chinese context. Control-oriented practice provides the premise for employees to pursue fairness and justice, commitment-based practice is in line with the “employee-centered” pursuit. Two kinds of practices mutually reinforce, ensuring the improvement of employee work well-being. Therefore, the person in charge or the CEO of the enterprise should rationally treat the human resource practices with these two different attributes, and in practice, they should comprehensively apply them to meet the needs of employees to improve their creativity. Second, enterprises should pay attention to employee work well-being experience, because work well-being is not only an important source of motivation for employee creativity, but also a key variable for human resource systems to improve employee creativity. In order to motivate employee creativity effectively, companies need to improve employee work well-being through implementing high-performance human resource practices that combine control and commitment practices. It can be seen that enterprise leaders should pay more attention to the improvement of employees’ work well-being and formulate policies and systems that are more conducive to improving work well-being. Third, although the human resource system is conducive to improve employee work well-being and stimulate their creativity, creativity is not only related to the employees’ own positive psychological state, but also influenced by transformational leadership. Therefore, while attaching importance to human resource management practices and employee work well-being, enterprises also need to pay attention to the influence of situational variables such as leadership style. So, leaders could try to change their leadership style or leadership traits to make them more responsive to employees’ needs for well-being and creativity.

### 5.3. Limitations and Future Research

Despite the fact that our study has some contributions, there are still several shortcomings. First, for sample, although we used various methods to try to contact the enterprises to participate in the questionnaire survey, due to the multiple time point research design, the geographical location of the company and the willingness of the employees, we only got 266 samples from 61 teams (the number of samples is insufficient), which may affect the accuracy of the overall judgment for Chinese companies. In addition, the enterprises that participated in our survey are companies that have close ties with the author and collaborators. Therefore, the sample is not collected strictly according to the method of “random sample”, and so some coefficients may have biased estimates. We suggest that it is necessary to continue to expand the sample size in future research and conduct a “random sample” survey if conditions permit.

Second, although we have proved that the employee work well-being has mediation effect between a high-performance work system and the employee creativity, the work well-being is only a sub-dimension of well-being. Some scholars pointed out that well-being should be comprised of life-well-being, work-well-being, and psychology well-being [17], yet we do not know whether these types of well-being could have the similar effects in strategic human resource management. Besides, employees’ work pressure is increasing currently, resulting in “work-family” conflict, which means the psychological intrusion of work on family life (people worry about their personal problems at work, but think about their work when they return home). Therefore, in fact, employees’ life-well-being, work-well-being, and psychology well-being are more difficult to distinguish. We suggest that, on the one hand, future research can carry out multi-dimensional research on well-being, on the other hand, we hope future research can use a more systematic perspective to explore the role of well-being in strategic human resource management.

Third, we incorporate transformational leadership as a moderation variable into the model. In the theoretical Hypothesis, we believe that transformational leadership will moderate the relationship between work well-being and employee creativity. However, we can find from the “Total Effect Moderation Model” test, that transformational leadership actually also moderate the positive impact of the high-performance work system on employee work well-being (*γ* difference = 0.113, *p* < 0.01). Thus, it is possible that the relationship between a high-performance work system and employee work well-being may be affected by some situational variables, such as leadership or organizational climate. We suggest in order to explore the formation mechanism of a high-performance work system on employee work well-being specifically, future research could pay attention to more situational variables when explore the role of work well-being in strategic human resource management.

Finally, we used a method that “supervisor-evaluation” to evaluate the employee creativity. However, this method only reflects the leader’s subjective assessment for the employee creativity, so its authenticity and accuracy are inevitably subject to error. Future research needs to adopt a more effective measurement method to evaluate the employee creativity, such as using questionnaire which combine “supervisor-assessment” with “self-assessment” or choosing some objective innovation indicators.

In addition, as far as the future research direction of a high-performance work system is concerned, we believe that there are three possibilities:

First, explore the impact of a high-performance work system on team-level variables, such as team performance, team innovation, etc. At present, there are many studies on the impact of a high-performance work system on organizational or individual-level variables, and the team is less concerned with the key elements of undertaking organizations and individuals, so it is possible to increase the relationship research between HPWS and team outcome variables in the future.

Second, focus on a high-performance work system on strategic goals. As the current environmental uncertainty increases and the high-performance work system evolves, future research should focus on a high-performance work system on the strategic goals of the characteristics, that is, to make the high-performance work system more refined and specific, such as research services-oriented high-performance work system [27], and flexibility-oriented high-performance work system [45].

Third, explore the antecedents of a high-performance work system. In the field of high-performance work system, the literatures often set a high-performance work system as an independent variable, instead, it ignores the exploration of a high-performance work system’s antecedent variables. Therefore, it is important to explore the antecedent of a high-performance work system.

## 6. Conclusions

Under the new economic normal, Chinese companies have begun to focus on innovation-driven development strategies, and they are particularly eager to enhance employee creativity. Based on the perspective of strategic human resource management, this paper finds that high-performance work system could significantly enhance employee creativity, and employees’ work well-being can play a mediating role in this process. In addition, transformational leadership can moderate the impact of employee work well-being on their creativity. Further, transformational leadership also moderates the indirect effects of high-performance work system on employee creativity.

## Figures and Tables

**Figure 1 ijerph-16-01640-f001:**
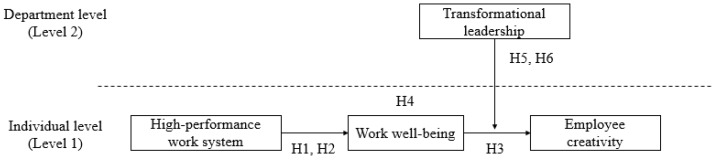
Research model.

**Figure 2 ijerph-16-01640-f002:**
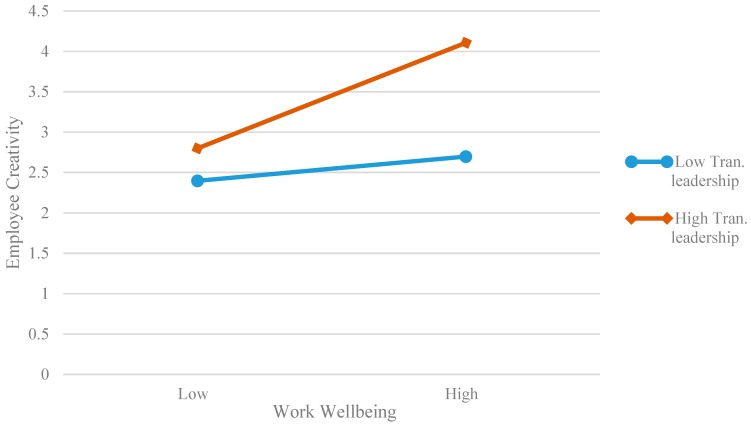
Moderation effect of transformational leadership on the relationship between Work well-being and employee creativity.

**Figure 3 ijerph-16-01640-f003:**
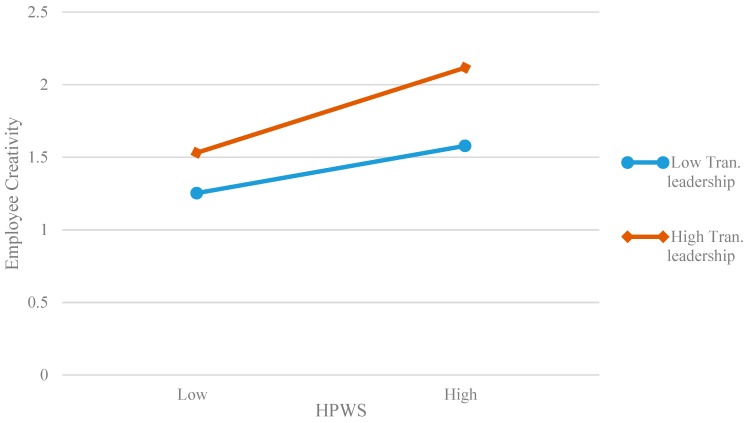
Moderation effect of transformational leadership on the indirect effect between a high-performance work system (HPWS) and employee creativity.

**Table 1 ijerph-16-01640-t001:** Descriptive statistics (N = 266).

Attribute		Frequency	Proportion (%)	Attribute		Frequency	Proportion (%)
Sex	Male	113	42.5%	Post	senior manager	10	3.8%
	Female	153	57.5%		middle manager	13	4.9%
Age	All	266	//		grassroots manager	19	7.1%
Working years	All	266	//		grassroots employee	224	84.2%
Education	below undergraduate	117	44%	Earners	high earners	93	35%
					low earners	173	65%
	undergraduate and above	149	56%	Owner	state-owned	87	32.7%
					private	179	67.3%

**Table 2 ijerph-16-01640-t002:** Confirmatory factor analysis results of concept discrimination validity.

MODEL	χ^2^	*df*	∆χ^2^	AIC	NNFI	CFI	RMSEA
**Four-Factor model** HPWS, WWB, TLS, EG	128.349	71	—	196.349	0.940	0.972	0.055
**Alternative Three-Factor model A** HPWS+TLS, WWB, EG	233.848	74	105.499 **	295.848	0.890	0.921	0.090
**Alternative Three-Factor model B** HPWS+WWB, TLS, EG	364.595	74	236.246 **	426.595	0.828	0.857	0.122
**Alternative Three-Factor model C** HPWS, TLS+WWB, EG	450.122	74	321.773 **	512.112	0.788	0.815	0.138
**Alternative Two-Factor** **model A** HPWS+TLS, WWB+EG	580.722	76	452.373 **	638.722	0.726	0.752	0.158
**Alternative Two-Factor** **model C** HPWS+EG, WWB+TLS	700.923	76	572.574 **	758.923	0.670	0.692	0.176
**Alternative Two-Factor** **model C** HPWS+WWB, TLS+EG	773.344	76	644.955 **	831.344	0.636	0.657	0.186
**Single-Factor model** HPWS+TLS+WWB+EG	964.574	77	836.225 **	1020.574	0.546	0.563	0.209

Note: HPWS indicates high performance work system; WWB indicates work well-being; TLS indicates transformational leadership; EG indicates employee creativity; AIC indicates akaike information criterion; NNFI indicates non-normed fit index; CFI indicates comparative fit index; RMSEA indicates root mean square error of approximation; + represents two factors to synthesize a variable; ** *p* < 0.01.

**Table 3 ijerph-16-01640-t003:** Mean, standard deviation, and correlations of variables ^a,b^.

Variables	M	SD	1	2	3	4	5	6	7	8	9
1 HPWS	4.768	0.756	0.878								
2 WWB	4.959	0.921	0.408 **	0.897							
3 TLS	5.286	0.833	0.544 **	0.441 **	0.957						
4 EC	4.868	0.916	0.373 **	0.469 **	0.338 **	0.950					
5 SEX	1.580	0.516	0.022	−0.073	0.009	−0.145 *					
6 AGE	29.410	6.316	−0.054	0.132 *	−0.071	−0.043	−0.292 **				
7 JAGE	6.746	6.205	−0.111	0.075	−0.093	0.038	−0.237 **	0.872 **			
8 EDU	1.560	0.497	−0.025	0.174 **	0.039	0.076	−0.232 **	0.193 **	0.095		
9 POST	1.429	0.496	−0.129 *	0.006	−0.179 **	0.057	−0.257 **	0.378 **	0.343 **	0.140 **	
10 INCOME	1.350	0.478	0.080	0.106	0.101	0.139 *	−0.202 **	0.171 **	0.109	0.284 **	0.241 **

Note: ^a^ N = 266, * means *p* < 0.05, ** means *p* < 0.01; High Performance Work System (HPWS), Work Well-being (WWB), Transformational leadership (TLS), Employee Creativity (EC); SEX means employee gender, males = 1 and females = 2; AGE indicates employee age; JAGE indicates the years of employee entry; EDU indicates employees’ educational level, specifically, the employees below undergraduate = 1, others = 2; POST means the position of the employee in the company, and the senior manager = 1, and the middle manager (department head) = 2, the grassroots manager (team leader) = 3, the grassroots employees = 4; INCOME indicates the employees’ income, the employees are low earners (below 8000 Yuan) = 1, the employees are high earners (over 8000 Yuan) = 2. ^b^ The correlation coefficient is in the lower triangle of the matrix; the diagonal line is the internal consistency coefficient α.

**Table 4 ijerph-16-01640-t004:** HLM results: mediating and moderation effect of work well-being.

Variables	Employee Creativity						Work Well-being	
	M1	M2	M3	M4	M5	M6	M7	M8
**Level-1**								
Sex	−0.198	−0.223	−0.186	−0.192	−0.193	−0.186	−0.071	−0.102
Age	−0.034 *	−0.028 *	−0.035 *	−0.033	−0.037 *	−0.033	0.020	0.005
Job age	0.026 *	0.025	0.028	0.029	0.029 *	0.027 *	−0.022	−0.011
Edu	−0.062	−0.008	−0.130	−0.069	−0.140	−0.148	0.143	0.275
Post	0.180	0.177	0.183	0.181	0.176	0.191	−0.026	0.031
Income	0.136	0.199	0.229	0.209	0.232 *	0.217 **	−0.025	−0.006
HPWS		0.457 **		0.288 **				0.498 **
WWB			0.401 **	0.318 **	0.404 **	0.402 **		
**Level-2**								
TLS					0.444 **	0.453 **		
WWB * TLS						0.252 **		
R^2^	0.091	0.197	0.205	0.284	0.266	0.327	0.082	0.173

Note: HLM means “Hierarchical linear Model”; All coefficient are normalized coefficients; N = 226, M is the model, * means *p* < 0.05, ** means *p* < 0.01; High Performance Work System (HPWS); Work well-being (WWB).

**Table 5 ijerph-16-01640-t005:** Indirect effect of work well-being.

Indirect Effect	Estimate	S.E.	Est./S.E.	*p*	95% CI
HPWS→WWB→EC	0.190	0.044	4.318	0.000	[0.116, 0.288]

Note: All coefficient are normalized coefficients; N = 266, High Performance Work System (HPWS); Work Well-being (WWB); Employee Creativity (EC).

**Table 6 ijerph-16-01640-t006:** Parameter estimation.

	a_05_	a_x5_	a_z5_	a_xz5_	*R^2^*	b_020_	b_x20_	b_m20_	b_z20_	b_xz20_	b_mz20_	*R^2^*
TLS	1.898	0.412 **	0.293 **	0.111 **	0.279	1.620	0.228 *	0.344 **	0.204 **	0.049	0.065	0.306

Note: N = 266, * means *p* < 0.05, ** means *p* < 0.01; Transformational Leadership (TLS); a_x5_, a_z5_, a_xz5_ are the non-standardized regression coefficients of Equation (1); b_x20_, b_m20_, b_z20_, b_xz20_, b_mz20_ are the non-standardized regression coefficients of Equation (2).

**Table 7 ijerph-16-01640-t007:** The results of moderated mediation (mediating variable is work well-being).

Moderation	Stage		Effect		
	First Stage	Second Stage	Direct Effect	Indirect Effect	Total Effect
	P_M,X_	P_Y,M_	P_Y,X_	P_M,X_ × P_Y,M_	P_Y,X +_ P_M,X_ × P_Y,M_
TLS					
Low (−1 s.d.)	0.356 **	0.311 **	0.203 **	0.111 **	0.314 **
High (+1 s.d.)	0.468 **	0.377 **	0.253 **	0.177 **	0.429 **
Difference	0.113 **	0.066 *	0.050	0.066 *	0.116 ^†^

Note: (1) † indicates *p* < 0.1,* indicates *p* < 0.05, ** indicates *p* < 0.01; (2) P_M,X_: path coefficient of HPWS and mediator variables; (3) P_Y,M_: path coefficient of mediator variable and employee creativity; (4) P_Y,X_: path coefficient of HPWS and employee creativity; (5) P_Y,X +_ P_M,X_ × P_Y,M_: total effect coefficient; (6) difference coefficient: high level coefficient−low level coefficient; (7) N =266.

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
