# Peer review of "High-Performance Work System, Work Well-Being, and Employee Creativity: Cross-Level Moderating Role of Transformational Leadership"

_ijerph, 2019, doi:10.3390/ijerph16091640_

Round 1

Reviewer 1 Report

Dear authors, 

The time involved in submitting your manuscript is greatly appreciated.

Despite this, the article presents a series of issues that must be noted and mended. The recommendations are presented separately by sections. Hopefully, they would be useful.

First, about all the paper, I would recommend avoiding a lot of abbreviations (HR, HPWS, AIC, M1, M2, M3, etc.). It is too hard for readers. Due that the Journal does not have any limitation in the number of words, please, try to reduce these abbreviations.  

Title: the title does not adequately reflect the content of the paper. For instance, the multilevel analysis is greatly appreciated, and it is not mentioned in the title. Please, adapt it to better inform the readers about that content.

Abstract:

Less information appears in the abstract. Maybe expanded by adding the most relevant findings.

Introduction:

Firstly, some of the references that you cite are too old. Even though the most relevant studies should be referenced, also the RECENT research must be included. Moreover, I recommend you to adapt the reference format to the Journal's suggestions instead of using APA's.

At the end of the literature review, the aims and the questions in the research should appear. Maybe to formulate the questions as hypothesis would be an option to clear this aspect. Another commentary, it is the possibility of including this part at the final of the introduction part; even a separate section could be a good option, in order to clear the final of the introduction and to serve as a connection with the method.

Method:

Please, try to better describe the sociodemographic data of your participants. In the same sense, give the readers with detailed information about the procedure for recruiting participants and collecting data.

Which Ethical committee approved the study protocol? Please, explain it.

Related to the instruments, please better inform about their psychometric quality and give to the readers some example of the items. If you can, please inform about previous studies where the same instrument has been used and the reliability obtained in that research. 

I think that the results section is very good, despite the use of abbreviations, that difficult the task for readers. 

Finally, a section related to limitations, future lines of investigations and the principal contributions of the research could be interesting. Suggestions for future research should be based not only on the limitations of the present research. Your paper has a lot of relevant implications for Human resource managers, for the society and policymakers, but you need to elaborate more on this topic.

Conclusion:

They don’t appear new conclusions on this part. This part does not add any new to the rest of the paper. Please, try to better condense your findings, or to highlight your main contribution to the field. 

Author Response

The first reviewer

The first reviewer's comments:

1. First, about all the paper, I would recommend avoiding a lot of abbreviations (HR, HPWS, AIC, M1, M2, M3, etc.). It is too hard for readers. Due that the Journal does not have any limitation in the number of words, please, try to reduce these abbreviations.  

2. Title: the title does not adequately reflect the content of the paper. For instance, the multilevel analysis is greatly appreciated, and it is not mentioned in the title. Please, adapt it to better inform the readers about that content.

3. Abstract:

Less information appears in the abstract. Maybe expanded by adding the most relevant findings.

4. Introduction:

Firstly, some of the references that you cite are too old. Even though the most relevant studies should be referenced, also the RECENT research must be included. Moreover, I recommend you to adapt the reference format to the Journal's suggestions instead of using APA's.

At the end of the literature review, the aims and the questions in the research should appear. Maybe to formulate the questions as hypothesis would be an option to clear this aspect. Another commentary, it is the possibility of including this part at the final of the introduction part; even a separate section could be a good option, in order to clear the final of the introduction and to serve as a connection with the method.

5. Method:

Please, try to better describe the sociodemographic data of your participants. In the same sense, give the readers with detailed information about the procedure for recruiting participants and collecting data.

Which Ethical committee approved the study protocol? Please, explain it.

Related to the instruments, please better inform about their psychometric quality and give to the readers some example of the items. If you can, please inform about previous studies where the same instrument has been used and the reliability obtained in that research. 

I think that the results section is very good, despite the use of abbreviations, that difficult the task for readers. 

Finally, a section related to limitations, future lines of investigations and the principal contributions of the research could be interesting. Suggestions for future research should be based not only on the limitations of the present research. Your paper has a lot of relevant implications for Human resource managers, for the society and policymakers, but you need to elaborate more on this topic.

6. Conclusion:

They don’t appear new conclusions on this part. This part does not add any new to the rest of the paper. Please, try to better condense your findings, or to highlight your main contribution to the field. 

The first response letter

Dear Prof.

Thank you very much for your letter and the comments from the referees about our paper submitted to IJERPH (ijerph-476777).  

We have learned much from the reviewer’ comments, which are fair, encouraging and constructive. After carefully studying the comments and your advice, we have made corresponding changes. Words in red are the changes I have made in the text, the following is the answers and revisions I have made in response to the reviewer' questions and suggestions on an item by item basis.

1. First of all, we agree with the reviewer's suggestion. We really wrote some professional words in abbreviated form. After this revision, we corrected them and changed them to a non-abbreviated form. For example, change HR to human resource; change HPWS for high-performance work systems; M1, M2... changed to Model 1, Model 2... In addition, in order to make the reader understand more clearly, we also explained professional statistical terms such as AIC, Rwg, ICC, etc.

2. Title: We agree with the reviewer's opinion that multilevel analysis, which is one of the highlights of this article, is indeed the main method used in our article. In order to enable readers to understand the content of the article more clearly through the topic, we will adjust the original topic “High-Performance Work System, Work Well-being, and Employee Creativity: The Moderating Role of Transformational Leadership” to "High-performance Work System, Work Well-being, and Employee Creativity: A cross-level study”

3. Abstract: We accept the recommendations of reviewers, and based on the content and findings of the article, we have expanded the abstract to make it more complete.

4. Introduction:

Thank you for your advice, we have deleted some old references in Introduction and also added some new references. In addition, we also have adjusted reference format according to IJERPH

We strongly accept the opinions of reviewer. In fact, we subtly summarized the questions in the last paragraph of the introduction. Therefore, we have made a clearer summary in the last paragraph of the introduction.

5. Method: In order to better describe the sociodemographic data of our participants, we combed the demographic variables in a table, so that readers can have a clearer understanding of the participants' data.

Based on the recommendation of the reviewer, we explained the process of collecting data in more detail. Specifically, “First, we determined the theoretical model by discussion and made up the questionnaire, and then we sent a message to the CEO or Human Resources Director of some large and medium-sized companies in China. We explain our research objectives and survey needs, some of companies responded to our emails and agreed to our request. Next, in these agreed companies, their CEO or HR Director helped us recommend some departments that are willing to participate in the research. It should be noted that all of our participants are voluntary. Our data collection took 2 months and passed 3 time points." Our project belongs to the general project of the National Social Science Fund of China and is a longitudinal project. Whether we can conduct a survey mainly depends on the opinions of the person in charge of the company we are contacting. It is gratifying that some CEOs or HR directors agreed to our request. We have explained in detail in section 3.1 of the article.

Based on the reviewer's comments, we made a more detailed description of the scales we use, including the items used and the quality of the measurements. We also added some studies that have used the same instruments in the past. For specific additions, we have identified in the 3.2 section of the article with a red handwriting

In addition, for the future research direction of high-performance work system, we have not only limited the content of this article, but also added three possibilities for future research. (1) Exploring the impact of high-performance work system on team-level variables, such as team performance, team innovation, etc.; (2) Focus high-performance work system on strategic goals; (3) Explore the antecedents of high-performance work system. The specific additions we have identified in the last paragraph of the article in red.

Besides, we also added some contents about relevant implications for Human resource managers, for the society and policymakers, supplemental content has been marked with red in 5.2.

6. Conclusions: We fully understand the opinion of the reviewer, so we have reduced the theoretical contribution of the article (5.1) to make it more prominent in our contribution to the field.

If you have any question about this paper, please don’t hesitate to contact us.  

Sincerely yours  

Reviewer 2 Report

Thank you for the opportunity to review your manuscript about “High-performance Work System, Work Well-being, and Employee Creativity: The Moderating Role of

Transformational Leadership”. HPWS and creativity, as far as I am concerned, are both interesting topics, likewise, the efforts to incorporate as a mediator in he linkage between HPWS and employee creativity were valuable. However, I do have several suggestions that I hope can help improve this research.

1. In the introduction part, the authors argued that no research has attempted to explore the formation mechanism of HR system and employee creativity from the perspective of employee work well-being so far.” However, to my knowledge, several Chinese scholars have explored the relationship between HPWS and employee, such as He et al. (2018) and Tang et al. (2017). These research can not be ignored. As you did not take these findings into account, I have to question the contributions of your research. This can be a fatal issue.

He, C., Gu, J., & Liu, H. (2018). How do department high‐performance work systems affect creative performance? a cross‐level approach. Asia Pacific Journal of Human Resources, 56(3), 402-426.

Tang, G., Yu, B., Cooke, F. L., & Chen, Y. (2017). High-performance work system and employee creativity: The roles of perceived organisational support and devolved management. Personnel Review, 46(7), 1318-1334.

2. The authors did not clearly answer the “why these variables?” questions. Overall, there didnt seem to be an overarching theory that explained the theoretical model and choice of variables. The authors suggested that you draw on JD-R theory, but I dont think they clearly explaine how this theory serves as the overarching theoretical framework that explain your full theoretical model and/or your inclusion of the specific variables you examined. As I was reading your paper, I found myself wondering how JD-R theory lead you to expect that employee well-being, in particular, mediates the relationship between HPWS and employee creativity.

Especially, your basement of mediating hypotheses is the argument the “response” hypothesis—job resources can improve employee motivation and job involvement under high-intensity job demands, which means that high-intensive job demands will motivate employees to make fully use of high-level job resources, and further work in a more positive state and achieve goals.” Whats the role of HPWS in your paper? Moreover, the “response” hypothesis is more likely to explain how job demands moderate the effect of job resource on employee motivation and job involvement.

Besides, there are so many styles of leadership, such as transactional leadership, humanity, servant leadership.... However, you do not explain clearly why transformational leadership is chosen as the specific moderator.

3. Theoretical explanations. Overall, I found the variables involved in the study to be interesting. However, I had a difficult time following some of the logic linking the relationships among variables.

4. Methodologically, I do like that the authors measured their variables from different sources. I do suggest that it would be better to use other methods (e.g., marker variable) to further test the common method bias. It is also suggested to adopt the bias-corrected bootstrapping procedure developed by Preacher and Hayes (2008) for mediation examination. Please add a part of analytical methods.

Podsakoff, P. M., MacKenzie, S. B., & Podsakoff, N. P. (2012). Sources of method bias in social science research and recommendations on how to control it. Annual review of psychology, 63, 539-569.

Preacher, K. J., & Hayes, A. F. (2008). Asymptotic and resampling strategies for assessing and comparing indirect effects in multiple mediator models. Behavior research methods, 40(3), 879-891.

5. Other issues. Please reformat tables. Please refer to those papers published in IJERPH and adjusted the format of references.

I hope that you will find these comments helpful as you continue to refine the manuscript and research these relationships.

Author Response

The second reviewer

The second reviewer's comments:

Thank you for the opportunity to review your manuscript about “High-performance Work System, Work Well-being, and Employee Creativity: The Moderating Role of

Transformational Leadership”. HPWS and creativity, as far as I am concerned, are both interesting topics, likewise, the efforts to incorporate as a mediator in the linkage between HPWS and employee creativity were valuable. However, I do have several suggestions that I hope can help improve this research.

1. In the introduction part, the authors argued that “ no research has attempted to explore the formation mechanism of HR system and employee creativity from the perspective of employee work well-being so far.” However, to my knowledge, several Chinese scholars have explored the relationship between HPWS and employee creativity, such as He et al. (2018) and Tang et al. (2017). These research can not be ignored. As you did not take these findings into account, I have to question the contributions of your research. This can be a fatal issue.

He, C., Gu, J., & Liu, H. (2018). How do department highperformance work systems affect creative performance? a crosslevel approach. Asia Pacific Journal of Human Resources, 56(3), 402-426.

Tang, G., Yu, B., Cooke, F. L., & Chen, Y. (2017). High-performance work system and employee creativity: The roles of perceived organisational support and devolved management. Personnel Review, 46(7), 1318-1334.

2. The authors did not clearly answer the “why these variables?” questions. Overall, there didn’t seem to be an overarching theory that explained the theoretical model and choice of variables. The authors suggested that you draw on JD-R theory, but I don’t think they clearly explaine how this theory serves as the overarching theoretical framework that explain your full theoretical model and/or your inclusion of the specific variables you examined. As I was reading your paper, I found myself wondering how JD-R theory lead you to expect that employee well-being, in particular, mediates the relationship between HPWS and employee creativity.

Especially, your basement of mediating hypotheses is the argument “the “response” hypothesis—job resources can improve employee motivation and job involvement under high-intensity job demands, which means that high-intensive job demands will motivate employees to make fully use of high-level job resources, and further work in a more positive state and achieve goals.” What’s the role of HPWS in your paper? Moreover, the “response” hypothesis is more likely to explain how job demands moderate the effect of job resource on employee motivation and job involvement.

Besides, there are so many styles of leadership, such as transactional leadership, humanity, servant leadership.... However, you do not explain clearly why transformational leadership is chosen as the specific moderator.

3. Theoretical explanations. Overall, I found the variables involved in the study to be interesting. However, I had a difficult time following some of the logic linking the relationships among variables.

4. Methodologically, I do like that the authors measured their variables from different sources. I do suggest that it would be better to use other methods (e.g., marker variable) to further test the common method bias. It is also suggested to adopt the bias-corrected bootstrapping procedure developed by Preacher and Hayes (2008) for mediation examination. Please add a part of analytical methods.

Podsakoff, P. M., MacKenzie, S. B., & Podsakoff, N. P. (2012). Sources of method bias in social science research and recommendations on how to control it. Annual review of psychology, 63, 539-569.

Preacher, K. J., & Hayes, A. F. (2008). Asymptotic and resampling strategies for assessing and comparing indirect effects in multiple mediator models. Behavior research methods, 40(3), 879-891.

5. Other issues. Please reformat tables. Please refer to those papers published in IJERPH and adjusted the format of references.

I hope that you will find these comments helpful as you continue to refine the manuscript and research these relationships.

The second response letter

Dear Prof.

Thank you very much for your letter and the comments from the referees about our paper submitted to IJERPH (ijerph-476777).  

We have learned much from the reviewer’s comments, which are fair, encouraging and constructive. After carefully studying the comments and your advice, we have made corresponding changes. Words in red are the changes I have made in the text, the following is the answers and revisions I have made in response to the reviewer' questions and suggestions on an item by item basis.

1. First of all, we are very sure of the reviewer’s questions about our article. For your first question, we think that what we mean is “no research has attempted to explore the formation mechanism of HR system and employee creativity from the perspective of employee work well-being so far In this sentence, we emphasize the role of employee work well-being in the process of high-performance work system affecting employee creativity, that is, we emphasize this mediation perspective (work well-being) instead of neglecting studies on the relationship between high-performance work system and employee creativity.

We don't want to ignore the two studies you listed. In fact, we also read these two studies in the writing process. Among them, He, Gu and Liu (2018) explain the relationship between high-performance work systems and employee innovation from OCB perspective, which means, they used OCB as the mediation variable. While Tang, Yu, Cooke, and Chen used perceived organizational support as an mediation mechanism to explain the relationship between high-performance work systems and employee creativity. Therefore, two studies did not use the perspective of employee work well-being as a process mechanism to explain the relationship between high-performance work systems and employee creativity. Therefore, we said that there is no study to explain the impact of high-performance work systems on employee creativity from the perspective of work well-being.

2. We appreciate the opinions of reviewer. First, we have explained in detail about why these variables were chosen in the introduction. Specifically, in the first paragraph of the introduction, we illustrated the need to study employee creativity—employee creativity is the foundation for organizational innovation and helps China build innovative countries. Second, in the second paragraph of the introduction, we elaborated why we chose high-performance work system? Mainly because there is less literature on the relationship between high-performance work system and employee creativity, and we believe that high-performance work system that represents organizational philosophy, systems, policies, and processes are more conducive to employee creativity. Third, in the third paragraph of the introduction, we explained in detail why the choice of employee work well-being as a mediator is because (1) work well-being is the ultimate pursuit of human beings; (2) there is no research in the past to explain the relationship between high-performance work system and employee creativity from the perspective of work well-being; (3) As far as the overall model is concerned, the logical that high-performance work system promotes the sense of well-being while further improving the employee creativity in line with the practical considerations that management research should take into account the interests of both parties. Finally, in the fourth paragraph of the introduction, we explained why the reason transformational Leadership: (1) comply with the perspective of strategic human resource management contingency theory; (2) In Chinese management context, the importance of leadership on employee attitudes and behavior; (3) the relationship between transformational leadership and creativity. We have already identified it in red in the introduction.

In addition, we fully understand the questions of reviewer, because we truly did not use an independent theoretical model to explain all variables. Our article actually used the JD-R model and the theory of creativity components to explain the mediating role of the model. For example, using the JD-R model to explain the relationship between high-performance work system and employees' work well-being, and then explaining the impact of work well-being on employee creativity based on the theory of creativity components. In fact, we initially thought about whether we can use an independent and complete theoretical framework to explain the model, but we did not find a suitable theory. We believe that there may be several reasons: (1)Theories in the field of strategic human resource management and organizational behavior focus more on employee capacity improvement or enterprise efficiency improvement, and there is not much attention to well-being theory. Therefore, we have not found a theory that can explain the effect of high-performance work system on well-being and creativity; (2) High-performance work system in the Chinese management situation is divided into “control-based” and “commitment-based”. Compared with Western management, the overall model is explained completely by a single theory.

We used “response” hypothesis in the JD-R model to explain the effect of high-performance work system on employee work well-being, but then used theory of creativity components to illustrate the relationship between work well-being and employee creativity. Therefore, we didn’t use a single theory to explain the mediation role of model, but the JD-R model and the theory of creativity components are used comprehensively.

In our article, the independent variable we used is employee-perceived HPWS, which is an employee’ overall perception of organizational philosophy, systems, practices, and processes. At the same time, according to Chinese traditional culture, when Chinese people complain their work, they are not often because of work stress (work demands) or less benefits, but more because of the uneven distribution of work results, so “response” hypothesis in JD- R model can respond to China's management practices. Although we believe that the work demands brought work pressure, but behind it is the Chinese people's most fancy fairness. Therefore, under the premise that fairness can be met, the pressure of work can be properly understood by employees. In other words, work demands and work resources are mutually reinforcing in the “response” hypothesis. In our model, it means that the work behind the demands shows a fair idea, in this circumstance, employees can make better use of work resources to complete the work, and thus promote the improvement of work well-being.

In addition, we also understand the question of reviewer on moderation variable—transformational leadership. We chose transformational leadership mainly because: (1) According to leadership theory, the relationship between transformational leadership and employee creativity is the most direct, And a lot of research uses transformational leadership as an independent variable to explore its positive relationship with employee creativity rather than transactional leadership or servant leadership; (2) According to the theory of creativity components, transformational leadership can promote employees' intrinsic motivation, professional knowledge and innovative thinking, and thus is closely related to creativity.

3. In order to give reviewer a clearer understanding of the logical relationship between our variables, we have rearranged the logical chain of the entire model: (1) HPWS—>employee creativity, we used AMO theory to prove; (2) HPWS —>work well-being, we used the “response” hypothesis in the JD-R model to explain; (3) work well-being—> employee creativity, we used the theory of creativity components to illustrate; (4) the moderate role of transformational leadership was explained by the "interactive point of creativity" point of view.

4. We really appreciate the rigor of the reviewer's measurement method, so we read the study by Podsakoff et al. (2012) based on the reviewer's opinion, then we based on the suggestions he has shown in article, we chose the specific method in his study (2003). For details, please refer to Section 4.3 of the article, we have been marked in red.

Podsakoff, N. P. (2003). Common method biases in behavioral research: A critical review of the literature and recommended remedies. Journal of applied psychology, 885, 879-903.

5. Thanks for the reminder of the reviewer, we have been reformat all tables and adjusted the format of references according to IJERPH.

If you have any question about this paper, please don’t hesitate to contact us.  

Sincerely yours  

Round 2

Reviewer 1 Report

Dear authors, 

many thanks for your responses. 

Reviewer 2 Report

I agree that the manuscript could be accepted after minor revision. Some grammar mistakes are still found here. For example, in the section of the abstract," the underlying mechanism and the boundary condition is not yet fully understood" the "is" should be replaced by "are". It is advisable for authors to read through the whole text and ask for language proof by English native speakers.